# Acridine Based *N*-Acylhydrazone Derivatives as Potential Anticancer Agents: Synthesis, Characterization and ctDNA/HSA Spectroscopic Binding Properties

**DOI:** 10.3390/molecules27092883

**Published:** 2022-04-30

**Authors:** Mária Vilková, Monika Hudáčová, Nikola Palušeková, Rastislav Jendželovský, Miroslav Almáši, Tibor Béres, Peter Fedoročko, Mária Kožurková

**Affiliations:** 1NMR Laboratory, Institute of Chemical Sciences, Faculty of Science, P. J. Šafárik University in Košice, Moyzesova 11, 041 67 Košice, Slovakia; maria.vilkova@upjs.sk; 2Department of Biochemistry, Institute of Chemical Sciences, Faculty of Science, P. J. Šafárik University in Košice, Moyzesova 11, 041 67 Košice, Slovakia; monika.hudacova@student.upjs.sk (M.H.); nikola.palusekova@upjs.sk (N.P.); 3Department of Cellular Biology, Institute of Biology, Faculty of Science, P. J. Šafárik University in Košice, Moyzesova 11, 041 67 Košice, Slovakia; rastislav.jendzelovsky@upjs.sk (R.J.); peter.fedorocko@upjs.sk (P.F.); 4Department of Inorganic Chemistry, Institute of Chemical Sciences, Faculty of Science, P. J. Šafárik University in Košice, Moyzesova 11, 041 67 Košice, Slovakia; miroslav.almasi@upjs.sk; 5Czech Advanced Technology and Research Institute, Centre of the Region Hana for Biotechnological and Agricultural Research, Palacký University, Šlechtitelů 241/27, 779 00 Olomouc, Czech Republic; tibor.beres@upol.cz

**Keywords:** acridine, benzohydrazide, ctDNA, HSA binding, spectroscopic study

## Abstract

A series of novel acridine *N*-acylhydrazone derivatives have been synthesized as potential topoisomerase I/II inhibitors, and their binding (calf thymus DNA—ctDNA and human serum albumin—HSA) and biological activities as potential anticancer agents on proliferation of A549 and CCD-18Co have been evaluated. The acridine-DNA complex **3b** (-F) displayed the highest *K_b_* value (*K_b_* = 3.18 × 10^3^ M^−1^). The HSA-derivatives interactions were studied by fluorescence quenching spectra. This method was used for the calculation of characteristic binding parameters. In the presence of warfarin, the binding constant values were found to decrease (K_SV_ = 2.26 M^−1^, *K_b_* = 2.54 M^−1^), suggesting that derivative **3a** could bind to HSA at Sudlow site I. The effect of tested derivatives on metabolic activity of A549 cells evaluated by the 3-(4,5-dimethylthiazol-2-yl)-2,5-diphenyltetrazolium bromide or MTT assay decreased as follows **3b**(-F) > **3a**(-H) > **3c**(-Cl) > **3d**(-Br). The derivatives **3c** and **3d** in vitro act as potential dual inhibitors of hTopo I and II with a partial effect on the metabolic activity of cancer cells A594. The acridine-benzohydrazides **3a** and **3c** reduced the clonogenic ability of A549 cells by 72% or 74%, respectively. The general results of the study suggest that the novel compounds show potential for future development as anticancer agents.

## 1. Introduction

Heterocyclic compounds, as the most important organic compounds, are frequently present in molecules of interest in medicinal chemistry [1,2,3,4,5]. Heterocycles demonstrate pharmacological activity via several mechanisms. Depending on the type of heteroatom present in the molecule, they may show various properties. Nitrogen heterocycles are among the most significant structural components of pharmaceuticals. They exhibit diverse biological activities and nitrogen heterocycles have always been attractive targets to synthetic organic chemists [6,7]. The prevalence of nitrogen heterocycles in biologically active compounds can be attributed to their stability and operational efficiency in human body and the fact of that the nitrogen atoms are readily bonded with DNA through hydrogen bonding [8]. In 2020, DiFranco and coworkers [9] demonstrated that exposure to a neo-synthetic bis(indolyl)thiazole alkaloid analog, nortopsentin 234, leads to an initial reduction of proliferative and clonogenic potential of colorectal cancer cells. Hamdy et al. [10] published the anti-apoptotic Bcl-2-inhibitory activity of synthesized 3-(6-substituted phenyl-[1,2,4]-triazolo[3,4-*b*]-[1,3,4]-thiadiazol-3-yl)-1*H*-indoles. Shao et al. [11] synthesized and tested aminoindazole derivatives as new irreversible inhibitors of wild-type and gatekeeper mutant FGFR_4_. Among the tested compounds, one aminoindazole exhibited excellent potency in both the biochemical and cellular assays, as well as modest in vivo antitumor efficacy. It is also worth mentioning the results obtained by Carbone et al. [12]. They synthesized a new series of thiazole derivatives and evaluated their ability to inhibit biofilm formation against the Gram-positive bacterial reference strain *Staphylococcus aureus* and Gram-negative strain *Pseudomonas aeruginosa*. The results showed that the new compounds affected the biofilm formation without any interference on microbial growth, and thus can be considered as promising lead compounds for the development of a new class of anti-virulence agents.

Acridine, a biologically active nitrogen-containing heteroaromatic ring, can be normally found in natural molecules. Todays, acridines are used as building blocks for the syntheses of heterocyclic systems, which have a strong influence on biological, pharmaceutical, and material sciences [13]. The biological activity of acridine derivatives is mostly due to the ability of the acridine moiety to intercalate between base pairs of double-stranded DNA through π-π interactions and interactions with topoisomerase I/II and telomerase. These compounds play an important role in the treatment of a variety of diseases, and they have been used clinically in past decades as antiviral, anticancer, anti-prion, antiprotozoal, anti-inflammatory, antineoplastic, and analgesic compounds [14,15,16,17,18,19,20]. In addition to several side effects, many drugs used in the past have increased resistance to them, resulting in a low therapeutic effect. These factors encouraged chemists to structurally modify acridine and produce different derivatives [21]. The structural modification of the acridine ring is the strategy of choice to improve the physicochemical and pharmacological properties.

*N*-Acylhydrazones, containing the -CO-NH-N= unit, have been the focus of interest for a long time due to their interesting properties, and they have found applications in medicine [22], agriculture [23] and materials engineering [24]. Recently, many compounds containing this moiety have been reported, which demonstrate that the introduction of this pharmacophore may result in high potential activities such as antivirus [25], antibacterials [26], antitumor [27], antileishmanial agents [28], etc. Acylhydrazone derivatives have also been widely used as ligands to prepare various complexes and chemosensors, which are always in the field of focus for researchers in materials due to their reversible acylhydrazone bond [27]. The structure-activity relationship (SAR) showed that the electron-withdrawing groups (-Cl, -NO_2_, -F, and -Br) were favoured in both DNA binding and anticancer activity, with the electron-donating groups (-OH and -OCH_3_) showing only moderate activity [29].

Our approach was to couple the acridin-4-yl and *N*-acylhydrazone moiety to obtain a new class of compounds, (acridin-4-yl)benzohydrazides. These compounds are not only a favorable basis to study structural effects (configurational and conformational isomerism) using nuclear magnetic resonance (NMR) spectral parameters, but also to study their interaction with calf thymus DNA and human serum albumin, too. Our study revealed a number of useful regularities in the ^1^H, ^13^C, and ^15^N NMR chemical shifts and coupling constants which depend on the conformation and configuration of *N*-acylhydrazones.

## 2. Results and Discussion

### 2.1. Chemistry

The novel compounds were synthesized using a procedure outlined in Figure 1 and Figure 2. The starting compound, acridin-4-carbaldehyde (**2**), was prepared according to the methods described in earlier literature [30,31]. Inspired by previously reported synthesis of 4-(bromomethyl)acridine [30], 4-(bromomethyl)acridine was reacted with the sodium salt of 2-nitropropane in dimethylsulfoxide at room temperature to afford the expected product acridine-4-carbaldehyde (**2**) in moderate yields. For the synthesis of benzohydrazides **1**, commercially available aryl aldehydes were converted to their methyl esters, which were reacted with excess hydrazine monohydrate (60%) under reflux to give the corresponding benzohydrazides **1**. Our approach to access substituted benzohydrazides **1a**–**e** is complementary to those already reported in the literature [32]. Aldehyde **2** was next allowed to react with a series of benzohydrazides **1** in ethanol providing compounds **3a**–**e** (Figure 1, Appendix A) in good yields.

(Acridine-4-yl)benzohydrazides **7b**–**d** were synthesized via a three-step reaction (Figure 2). The derivatives of series **7** were prepared from aldehyde **2**. By using conditions close to those of lit. [33] for the oxidation of aldehyde **2** (2.90 mmol of aldehyde **2**, 4.63 mmol of iodine in 8 mL methanol, 8.11 mmol of KOH in 8 mL of methanol), but with an extended reaction time (2 h at 0 °C and 4 h at room temperature (rt)) in order to allow the reaction to go to completion, aldehyde **2** was oxidized with alkaline iodine to directly lead to methyl acridine-4-carboxylate (**4**) isolated in 73–91% yield. Aridine-4-carboxylate (**4**) was treated by hydrazine hydrate (22.25 mmol) at the reflux temperature of ethanol until the disappearance of the starting material. The expected acridine-4-carbohydrazide (**5**) was isolated in 75% yield [34]. To synthesize hydrazides **7b**–**d**, we simply treated **5** with aryl aldehydes **6b**–**d** in ethanol as before and obtained the expected [(acridin-4-yl)methylidene]benzohydrazides **7b**–**d** in 80–90% yields (Figure 2, Appendix A).

The structural characterization of the synthesized compounds was performed using a combination of 1D and 2D NMR techniques. Hydrazones are suitable compounds for examining the stereospecificity of NMR parameters. Their configuration and conformation can be determined by NMR spectroscopy, as evidenced by a number of published scientific articles [35]. It is well known that hydrazones can exist as geometrical isomers due to the C=N double bond, but C(O)-N and N-N bonds also allow them to exist as conformers [36]. In the ^1^H, ^13^C, and ^15^N NMR spectra for derivatives **3a**–**e** measured in DMSO-d_6_, only a single set of signals was present (Appendix A and NMR spectra in Appendix A), while in the ^1^H, ^13^C and ^15^N NMR spectra for derivatives **7b**–**d** measured in acetone-d_6_, the signals were duplicated (Appendix A and NMR spectra in Appendix A). The geometric configuration of the C4=N3 double bond was determined based on the stereospecificity of the heteronuclear one bond spin–spin coupling constant ^1^*J*_CH_ coupling constants with respect to the orientation of the nitrogen lone pair, the ^1^*J*_CH_ coupling constant with anti-orientation of the relevant C-H bond to the nitrogen lone pair being about 160–170 Hz [37]. Heteronuclear one bond spin–spin coupling constant ^1^*J*_C4H4_ (167.4–168.6 Hz for **3a**–**e** and ^1^*J*_C4H4_ = 162.6 Hz for compounds **7b**–**d**, Figure 1, Appendix A) indicated that the C4=N3 double bond existed in the *E*-configuration in derivatives **3a**–**e** and **7b**–**d** [21]. The determination of the *Z*_C(O)-N2_ conformation was performed based on the heteronuclear two bond spin–spin coupling constants (^2^*J*_C1H2_ = 10.8 Hz for **3a** and 11.4 Hz for **3c**) and the NOESY enhancements from 1D NOESY spectra measured in DMSO-d_6_ for derivative **3a** (see Appendix A) and 2D NOESY spectra measured in acetone-d_6_ for derivative **7c**. The duplicate signals in the NMR spectra of (acridine-4-yl)benzohydrazides **7b**–**d** can be attributed to the existence of *E*_N-N_/*Z*_N-N_ conformers. The hydrogen bond N10′···H2 produce a low-field shift of the H2 lines to 15.00 ppm and an up-field shift of the N10′ (from −95.9 to −96.3 ppm; see Appendix A). This downfield shift is a result of a decrease in the electron density around the hydrogen nucleus and the deshielding effect from the electronic currents of the acceptor atom. While this deshielding effect is experienced by the donor nucleus, the chemical shift of the acceptor nucleus moves to a lower frequency due to an overall increase in electronic shielding. The redistribution of electron densities upon the formation of hydrogen bonds gives rise to observable changes in the scalar couplings between nuclei associated with hydrogen bonds [38]. Hydrogen bond N10′···H2 formation also resulted in a decrease in the ^1^*J*_N2H2_ coupling constant and a corresponding decrease in the strength of the N2-H2 bond. Similarly, a decrease in the ^1^*J*_C4H4_ constant of derivative **7** in comparison with compound **3** is associated with the presence of the hydrogen bond C(O)···H4, which also stabilizes the *Z*_N-N_ form of derivative **7** (Figure 1, Appendix A and NMR spectra in Appendix A). The second form of derivative **7** is the *E*_N-N_ conformer. In addition to the results of the NMR spectra studies, the structures of both conformers in derivatives **7** were corroborated by the NOESY data. A NOESY cross peak was detected between acridine proton H-3′ and proton H-4 in the *Z*_N-N_ conformer, while NOESY cross-peaks were also recorded between protons H-2 and H-4 and protons H-2/H-4 and H-5′. An additional argument for the existence of *E*_N-N_/*Z*_N-N_ conformers are the presence of chemical shifts which are almost identical for both conformers. Interestingly, no significant preference was observed for one of the *Z*_N-N_/*E*_N-N_ conformers in the case of derivative **7c** (*Z*_N-N_/*E*_N-N_, 10:12), but a small preference for the *Z*_N-N_ conformer was observed for derivatives **7b** and **7d** (*Z*_N-N_/*E*_N-N_, 10:7 (**7b**), 10:5 (**7d**), Figure 1).

The infrared spectra (IR) of selected compounds (**3a**–**d** and **7b**–**d**) and the assignment of characteristic absorption bands with corresponding wavenumbers (in cm^−1^) are listed in Appendix A (IR spectra in Appendix A). The presence of acridine and phenyl moieties in compounds **3a**–**d** and **7b**–**d** are evident from several stretching vibrations of aromatic C-H (*ν*(CH)_ar_) and C=C (*ν*(C=C)_ar_) located in the range 3089–3010 cm^−1^ and 1603–1506 cm^−1^, respectively. The presence of these components is also confirmed by the scissoring (*γ*(CCH)_ar_) and out-of-plane (*δ*(CCH)_ar_) vibrations of CCH which occur in the IR spectra of the compounds at about 750 cm^−1^ and 1020 cm^−1^. The successful preparation of the hydrazides through a condensation reaction of carbohydrazide and aldehyde derivatives was confirmed by the presence of a stretching vibration of the secondary amine in the range of 3190–3369 cm^−1^. The wide range of wavelengths observed in these results can be explained by the involvement of the amine group in the hydrogen bond system in solid state. The presence of a hydrazide bond in the case of compounds **3a**–**d** and **7d** was also confirmed by the presence of an azomethine vibration (*ν*(C=N)) at approximately 1620 cm^−1^ [39]. Moreover, the presence of the carbohydrazide group is also confirmed by a characteristic intense absorption band of about 1650 cm^−1^ in the IR spectra of all compounds and belongs to the stretching vibration of the carbonyl group (*ν*(C=O)). The IR spectroscopy results are in good agreement with the molecular structure of the synthesized compounds and NMR spectroscopy measurements.

### 2.2. HSA Binding Experiments

#### 2.2.1. Fluorescence Quenching Properties

Human serum albumin (HSA) emission spectra were recorded in the absence and presence of different amounts of acridine derivatives (**3a**–**3d**) at a range of 285 to 550 nm upon excitation at 280 nm. The emission maximum of fluorescence intensity of HSA was identified at 340 nm. The presence of derivatives **3a**–**3d** caused a concentration-dependent quenching of HSA fluorescence with a moderate change in the emission maximum which did not alter the shape of the peak (Figure 2 and Appendix A). The fluorescence intensity of HSA in the presence of the acridine derivatives decreases in the relation **3d** < **3b** < **3c** < **3a**, with decreases of 58%, 51%, 43% and 38%, respectively. Moreover, the reduction of fluorescence emission of HSA by derivatives **3a**–**3d** was accompanied by an apparent change in the position of the maximum wavelength of the fluorescence emission, with a blue shift of 6 nm (**3d**) ≈ (**3b**) < 7 nm (**3c**) < 8 nm (**3a**). The observation of a significant blue shift upon the addition of derivatives **3a**–**3d** suggests an increased hydrophobicity of the region near the Trp residues in the presence of the derivatives [40]. These results indicate that the interaction between acridine derivatives and HSA could lead to a change in the secondary structure of the protein, thereby causing changes in HSA in the environment around the Trp residues [41]. A major fluorescence residue of HSA is Trp (Trp 214) located in the binding domain IIA (site I), and the quenching of the fluorescence emission intensity of HSA upon increasing amounts of acridine derivatives suggests the presence of Trp residues of HSA at or near the binding site with the acridine derivatives [42].

The appearance of an isoactinic point at 430 nm might also indicate the existence of an equilibrium between bound and free drugs, with the presence of such an equilibrium possibly emphasizing the formation of the drug–protein complex [43]. Fluorescence quenching can be divided into the two processes of dynamic and static quenching. In dynamic quenching, the fluorophore and quencher are in contact in an excited state, increasing the temperature and resulting in faster diffusion and a higher frequency of collision, which in turn increases the quenching constant [44]. In comparison, static quenching is typical in complexes formed between the fluorophore and quencher in a ground state in which the increasing temperature weakens the stability of the complex [45]. The acridine derivatives and HSA was evaluated at different temperatures (25, 30 and 35 °C), as is shown in Appendix A. Quantitative analysis was performed by using the Stern–Volmer Equation (1):(1)$$\frac{F_{0}}{F}=1+K_{SV}\left[Q\right]=1+K_{q}\tau_{0}\left[Q\right],$$
where *F*_0_ and *F* represent the fluorescence intensity of HSA in the absence and presence of acridine derivatives, respectively. *K*_SV_ is the Stern–Volmer dynamic quenching constant which was determined from the plot of relative intensity *F*_0_/*F* of fluorescence vs. the concentration of the acridine derivative. *K*_q_ is the quenching rate constant of biomolecules, which is known to be around 2.0 × 10^10^ M^−1^ s^−1^, and τ_0_ is the average lifetime of the fluorophore in the absence of the quencher, with a typical value equal to around 10^−8^ for biomolecules [46]. The calculated values of *K*_SV_ and *K*_q_ at different temperatures are listed in Table 1. The quenching constant *K*_SV_ of derivatives **3a**–**3d** were of 10^5^ M^−1^. The *K*_SV_ values recorded for derivatives **3a** and **3b** decreased at increasing temperatures, while those of derivatives **3c** and **3d** increased at increasing temperatures. The increased *K*_SV_ values at increasing temperature in the presence of derivative **3c** may indicate that the binding forces are mainly hydrophobic (endothermic apolar interactions are strengthened at increasing temperatures) [45]. Additionally, the *K*_q_ values obtained for HSA of 10^13^ M^−1^ s^−1^ are greater than the limiting diffusion rate constant of diffusional quenching for biopolymers (2.0 × 10^10^ M^−1^ s^−1^). This would suggest that the observed fluorescence quenching process of the HSA emission by derivatives **3a**–**3d** is not initiated by the dynamic process, but instead by a static process with ground-state complex formation.

In order to evaluate the magnitude of the interaction, other parameters such as the binding constant ( *K_b_*) and the number of binding sites (n) on the drug–HSA complex were calculated at three temperatures using Equation (2) [47]:(2)$$log\left(\frac{F_{0}-F}{F}\right)=logK_{b}+nlog\left[Q\right],$$
where *F*_0_ and *F* are the fluorescence intensity of the fluorophore (HSA) in the absence and presence of the quencher-acridine derivative, and [*Q*] is the quencher concentration. The values of *K_b_* and *n* were determined using the linear regression of a plot of *log*(*F*_0_ − *F/F*) to *log*[*Q*] (Appendix A). The values of *K**_b_* and n at the three temperatures are summarized in Table 2. The results show that values of n are almost equal to 1 at increasing temperatures, a finding which indicates the existence of a single binding site. Since the values of *K_b_* were 10^5^ M^−1^ (**3a**–**3c**) or 10^4^ M^−1^ (**3d**), it would therefore be suitable for distribution in the plasma in vivo.

The interaction of the quencher derivatives **3a**, **3b** and **3d** with HSA are accompanied by a decrease in *K_b_* values and n values at increasing temperatures, a trend which indicates the destabilization of the drug-HSA complex under these conditions. In addition, Table 2 also showed that the values of *K**_b_* decreased at increasing temperatures, indicating that the interaction process also involves a static quenching mechanism [42]. In the case of acridine derivative **3c**, the values of *K**_b_* and *n* increased at increasing temperatures resulting in an endothermic reaction and increment of the stability of drug-HSA complex, a process which also implies that the ability of the derivatives to bind to HSA was enhanced at increasing temperatures [40,42]. The findings suggest that derivative **3c** could be delivered by HSA in vivo more effectively than derivatives **3a**, **3b**, and **3d**.

Based on the *K_b_* dependency on temperature, it is, therefore, possible to analyse the temperature-dependent thermodynamic parameters which can be considered as responsible for the formation of a complex [42]. The main thermodynamic parameters which are linked to the binding of small molecules to biomacromolecules, such as enthalpy change (Δ*H*) and entropy change (Δ*S*), were calculated using the van’t Hoff Equation (3):(3)$$logK_{b}=-\frac{\Delta H}{\left(2.303RT\right)}+\frac{\Delta S}{\left(2.303R\right)},$$
where *K**_b_* is the binding constant, *R* is the gas constant (8.314 J.mol^−1^. K^−1^), and T is the experimental temperature in Kelvin. The values of Δ*H* and Δ*S* were evaluated from the slope and intercept of the van’t Hoff plot. The relationship between the values of Δ*H* and Δ*S* can provide additional information about the primary forces of the interaction between the small molecules and the macromolecules. Thus, if Δ*H* > 0 and Δ*S* > 0, a hydrophobic interaction has occurred, while if Δ*H* < 0 and Δ*S* < 0, the main binding forces are hydrogen bonding and van der Waals forces [46]. The Gibbs free energy change (Δ*G*) was calculated using the Gibbs-Hemholz Equation (4):(4)ΔG=ΔH−TΔS

The calculated values of the thermodynamic parameters Δ*H*, Δ*S* and Δ*G* are summarized in Table 2. The thermodynamic profile of the interaction of compounds **3a**–**3d** with HSA was created from the calculated thermodynamic parameters. The negative values of Δ*G* suggest that the interaction between acridine derivatives **3a**–**3d** and HSA is spontaneous, while the negative values of Δ*S* and Δ*H* indicate that derivatives **3a**, **3b** and **3d** primarily bind with HSA through hydrogen bonding and van der Waals forces [43]. The interaction of compound **3c** with HSA is driven by the positive values of Δ*S* and Δ*H*, but the unfavourable positive values of Δ*H* for the spontaneous binding process are compensated for by the positive value for the entropy change that supports the negative values of the Gibbs free energy. In addition, the binding process is entropically driven (endothermic) with hydrophobic interaction emerging as the main intermolecular force in the interaction of compound **3c** with HSA [44,48,49,50].

#### 2.2.2. Effect of Compounds **3a**–**3d** on HSA Conformation

Synchronous emission spectroscopy is fluorescence spectroscopy performed at a constant wavelength, and the technique to provide information on the effect of the binding of small molecules on the microenvironment in the vicinity of tryptophan (Trp) and tyrosine (Tyr) as chromophore residues of protein [51]. Synchronous fluorescence spectra assay simultaneously records the excitation and emission monochromators at a constant wavelength interval; the characteristic spectra of Tyr and Trp residues are found at Δλ = 15 or 60 nm [48,51]. The shift in the position of fluorescence emission maximum corresponds to changes in the polarity around the chromophore molecule [52]. Specifically, a red shift indicates the increment of hydrophilicity, and a blue shift reveals the increment of hydrophobicity around the fluorophores of serum albumins [48,51]. As is shown in Figure 3 and Appendix A, the fluorescence intensity of the synchronous emission spectra of HSA was found to decrease with increasing amounts of acridine derivatives **3a**–**3d**, a result which further demonstrates the occurrence of fluorescence quenching in the binding process. In synchronous emission spectra, a reduction in fluorescence intensity without any shift implies that no disturbance has occurred in the microenvironment around the particular residue. The synchronous emission quenching for HSA due to the action of the quencher implies that it is most probably located adjacent to the Trp and Tyr residues [49]. The rate of emission quenching for HSA at Δλ = 15 nm or 60 nm was compared by graphical dependency F/F_0_ vs. [quencher] (Figure 3, Appendix A and Appendix A). The results show that changes in fluorescence quenching intensity occurred more frequently around Trp residues (Δλ = 60 nm) than those of Tyr (Δλ = 15 nm). This would indicate that Tyr contributes more to the quenching of the intrinsic fluorescence of HSA at the excitation wavelength of 280 nm [50]. The evidently higher rate reduces the fluorescence intensity around Trp, further indicating that the binding site is site I in the IIA HSA subdomain.

A useful method for studying drug-HSA interactions is three-dimensional fluorescence spectroscopy, which can offer information about the structural changes to the polypeptide backbone structure and the microenvironment polarity around the Trp and Tyr residues [49,50,51,53,54,55,56,57,58,59]. The results of 3D fluorescence are shown in Figure 4 and Appendix A. Peak a (λ_ex_ = λ_em_) is second-order scattering, while peak b (2λ_ex_ = λ_em_) is Rayleigh scattering. The change at peak 1 (280/339 nm) reflects a change in the polarity of the microenvironment around the Trp and Tyr residues, and peak 2 (230/336 nm) characterizes the polypeptide backbone structure [59]. The results show a significant blue shift for both peak 1 (8 nm) and peak 2 (10 nm) in the presence of compound **3a**, and a moderate blue shift for peak 1 (1–4 nm) and peak 2 (4–8 nm) in the presence of compounds **3b**–**3d**. These findings indicate that binding with derivatives **3a**–**3d** causes a partial change in the conformation of HSA and led to a decrease in the polarity surrounding the Trp and Tyr residues [58]. The fluorescence intensities of both peaks decreased after the addition of derivatives **3a**–**3d**, with a more pronounced decrease observed in the fluorescence intensity of peak 2. The changes in the peaks decreased in the following relations: peak 1: **3a** > **3c** > **3d** > **3b;** and for peak 2: **3b** > **3c** > **3d** > **3b**. The changes observed in the 3D spectra correlate with the results of earlier quenching fluorescence experiments. The decreased intensity of peaks 1 and 2 may be due to the increased exposure of some previously buried hydrophobic regions and may reflect the π–π* transition of fluorophores and the π–π* transition of the C=O bond, respectively [59]. The results also suggest that some environmental and conformational changes to HSA may have occurred upon the addition of the acridine derivatives (Figure 4 and Appendix A).

#### 2.2.3. Determining the Binding Site of Acridine Derivatives **3a**–**3d** on the HSA Molecule

Displacement experiments were performed using warfarin and ibuprofen as site marker fluorescence probes, as these agents have been proven to bind with HSA at Sudlow sites I and II, respectively [58]. The fluorescence quenching spectra of HSA:acridine derivatives **3a** (1:1) (**3b**–**3d**) (Figure 5) in the presence of increasing concentrations of specific site markers, warfarin or ibuprofen, were studied.

Interestingly, the addition of warfarin led to a significant quenching of the intrinsic fluorescence maximum of the HSA:acridine complex with a red shift. The red shift can be explained by the increasing polarity of the region surrounding the Trp site [59]. The spectra show a new fluorescence peak maximum at around 375 nm. In the presence of ibuprofen, there was no change in the emission spectra of the HSA:acridine complex (**3a**–**3d**).

The change in the emission spectra of the HSA:acridine complex in the presence of warfarin and ibuprofen was analysed using Equation (5).
(5)$$I\left(\%\right)=\frac{F_{2}}{F_{1}}\times 100\%,$$
where *I* is the percentage of the initial fluorescence, and *F*_1_ and *F*_2_ are the fluorescence intensity of the HSA:acridine complex in the presence and absence of the site marker [60]. The results show that warfarin caused a significant decrease in the fluorescence intensity of the HSA-acridine complex, while ibuprofen caused only a minimal change in intensity. These results indicate that warfarin competes with the acridine derivatives for binding sites on HSA at Sudlow site I in the IIA HSA subdomain.

In addition to the spectra assays, reverse titration studies were also performed. The intensity of the fluorescence of the HSA:site marker complex was recorded at increasing concentrations of acridine derivative **3a**, with the obtained results given in Appendix A. Interestingly, the titration of varying concentrations of **3a** with both the HSA-ibuprofen and HSA-warfarin complexes showed a quenching of fluorescence emission (Appendix A). Taken together, these results suggest that the binding between the site marker and HSA is affected by derivative **3a** in both cases. The obtained data were analysed using Equations (3) and (4). The linear Stern-Volmer dependence and dependence *log*(*F*_0_ − *F*)/*F* = *log*[*Q*] was used to obtain Stern-Volmer quenching constant values (*K*_SV_) and apparent binding constant values (*K**_b_*). Previous studies have shown that if a drug has the potential to bind to the same site as the marker, the interaction of this drug with HSA shows an apparent reduction in the *K**_b_* value [59]. The values of *K*_SV_ and *K**_b_* for the HSA-**3a** complex were found to be 2.67 × 10^5^ M^−1^ and 6.10 × 10^5^ M^−1^. In the presence of ibuprofen, there was an increase in the experimental values: *K*_SV_ = 3.28 M^−1^ and *K**_b_* = 6.67 M^−1^ were calculated. A marked change in the quenching constant was also observed after the addition of ibuprofen, a result which could be partly due to the interaction between derivative **3a** and ibuprofen. Similar changes and associated interactions between the marker and the studied ligand have been observed in other studies in which competitive experiments were conducted [58]. In the presence of warfarin, the values of the constants were found to have decreased: *K**_SV_* = 2.26 M^−1^ and *K**_b_* = 2.54 M^−1^. Derivative **3a** competes with warfarin on HSA, which suggests that derivative **3a** could bind to HSA at Sudlow site I. According to the obtained results, it is possible to assume that derivative **3a** binds to HSA at the same site as warfarin.

### 2.3. DNA Binding Properties

Changes in absorbance intensity (either hyperchromic or hypochromic) and a shift in the peak position (hypsochromicity or batochromicity) are typical spectral properties closely related to the interaction of a drug with DNA [60,61]. Figure 6 shows the UV-Vis spectra of acridine derivatives **3a**–**3d** in both the absence and presence of varying concentrations of ctDNA. In free form, derivatives **3a**–**3d** displayed absorption bands at 300–500 nm with a maximum at around 400 nm. The absorption spectra of compounds **3a**–**3d** in the presence of increasing concentrations of ctDNA show hypochromicity (16–51%) and a slight hypsochromic shift (Δλ = 1–9 nm, **3a**–**3d**), findings which indicate that the acridine derivatives have interacted with DNA (Table 3). The hypochromic effect showed a decrease in the following relation: **3b** > **3a** > **3c** ≈ **3d**, with derivatives **3b** and **3a** displaying the most notable hypsochromic shifts. Hypochromism typically occurs as a result of the contraction of ctDNA in the helix axis, and this conformation change is often a result of a surface binding complex developing between the small molecules and the DNA, either through external contact or in the case of strong interaction between the small molecules and DNA [62,63]. Classical intercalating compounds are able to couple their π orbital with the π of the base pair of DNA during intercalation into DNA, thereby decreasing the π → π* transition energies [64] and inducing hypochromism combined with a batochromic shift [65]. In contrast, electrostatic interaction can also show hyperchromism, an effect that indicates the increasing likelihood of electrons in the π–π* transition of the extended resonance system. This process is usually accompanied by a hypsochromic shift as a response to the increasing electron density in π-orbitals; it also stabilizes the orbitals and results in an increase in the energy gap between the π and π* orbitals [18,66]. On this basis, it is possible to state with confidence that the observed changes in the absorption spectra of acridine derivatives **3a**–**3d** in the presence ctDNA are a result of the electron-rich DNA donating electrons to the π-orbitals of the system. Therefore, our results suggest that acridine derivatives **3a**–**3d** had all interacted with ctDNA.

The binding constant *K_b_* of compounds **3a**–**3d** was determined using the modified Benesi-Hildebrand Equation (6) [51]:(6)$$\frac{A_{0}}{A-A_{0}}=\frac{\varepsilon_{B}}{\varepsilon_{B\;+\;\mathrm{DNA}}}+\frac{\varepsilon_{B}}{\varepsilon_{B\;+\;\mathrm{DNA}}}\;\times \;\frac{1}{K_{b}\left[\mathrm{DNA}\right]},$$
where A_0_ and A are the absorbances of the acridine derivatives in the absence and presence of ctDNA; ε_B_ and ε_B_ + DNA are the molar extinction coefficients of the acridine derivatives individually and in the bound complex, respectively; *K_b_* is the binding constant; and [DNA] is the concentration of ctDNA. The binding constant *K_b_* was estimated from the intercept-to-slope ratios of A_0_/(A − A_0_) vs. 1/[DNA] plots (insert graph Figure 6). The values of binding constant for the interaction of the acridine derivatives with ctDNA increased in the following relation: **3d** < **3c** < **3a** < **3b**. The highest *K* value recorded from the series of (acridin-4-yl)benzohydrazide compounds was that of the fluoro-substituted derivative **3b** (*K_b_* = 3.18 × 10^3^ M^−1^), a result which is in accordance with other recent studies. Accordingly, the *K* values obtained for acridine derivatives **3a**–**3d** were in the range 1.01–3.18 × 10^3^ M^−1^, since previous research has reported *K_b_* values from 10^4^ to 10^6^ M^−1^ for intercalation complexes, binding constants which are conspicuously smaller than those of groove binders (10^5^–10^9^ M^−1^) [67]. In addition, acridine derivatives **3a**–**3d** displayed a hypochromic effect in combination with a hypsochromic shift in the presence of ctDNA, a feature which differs from that of classical intercalation, which causes the combination of hypochromic and bathochromic shifts. It is therefore possible to suggest that the studied derivatives bind to DNA through partial slight intercalation or by groove binding via the acridine scaffold through a combination of other external binding such as electrostatic interaction with another part of the scaffold such as, for example, the benzohydrazone linker.

### 2.4. Biological Evaluation

#### 2.4.1. The Effect of **3a**–**3d** on Metabolic Activity of A549 Cancer Cell Line and CCD-18Co Fibroblasts Evaluated by MTT Assay

The effect of acridine-benzohydrazides (**3a**–**3d**) on the metabolic activity of the lung adenocarcinoma cell line (A549) and the normal colon fibroblast cell line (CCD-18Co) was evaluated using a 3-(4,5-dimethylthiazol-2-yl)-2,5-diphenyltertrazolium bromide (MTT) cell assay. This assay is based on the metabolization of yellow MTT to purple formazan crystal by mitochondrial dehydrogenases, and therefore only metabolic active cells are capable of that reaction [68]. The CCD-18Co cell line was used as a control to check for derivatives selectivity. The doxorubicin belongs to a group of anthracyclines, which represent the chemotherapeutic regiments for the treatment of NSCLC (non-small cell lung carcinoma) [47,69] even today, and acts as antineoplastic mainly by interaction with the DNA topoisomerase II and by creating DNA single- and double-strand breaks [47]. Previous studies described detailed cytotoxic effects of doxorubicin in the human lung adenocarcinoma A549 cell line [29,52,70,71,72]. Doxorubicin resembles our compounds in its biological action; therefore, it was employed as a positive control. As shown in Figure 7, metabolic activity of A549 cells significantly decreased after 24 h (Figure 7A) and 48 h (Figure 7B) treatment with acridine-benzohydrazides **3a**–**3d** (*p* < 0.01, *p* < 0.001) in concentrations of 50 and 75 μM, respectively.

Similarly, in the case of CCD-18Co fibroblasts, we observed a significant loss of metabolic activity after 48 h treatment with acridine-benzohydrazides **3a**–**3c** (*p* < 0.001) in the concentration range 25–75 μM (Figure 8). Doxorubicin in the full concentration range 5–75 μM (*p* < 0.001) significantly decreased the metabolic activity of A549 and CCD-18Co cells. Dimethyl sulfoxide (DMSO) was used as solvent control, which did not show any significant toxicity against A549 cells in comparison with CCD-18Co cells, where we observed a considerable decrease of metabolic activity in concentration 0.25% (*p* < 0.01), 0.50% and 0.75% (*p* < 0.001).

Based on the results from the MTT assay, the IC_50_ values for all compounds were evaluated. As shown in Table 4, the IC_50_ values for A549 cells after 24 h treatment was >75 μM, except derivative **3c** (IC_50_ = 73 μM). After 48 h incubation, the IC_50_ values were in the range 37–62 μM. The effect of tested derivatives on metabolic activity of A549 cells evaluated by MTT assay (48 h) decreased as follows **3b**(-F) > **3a**(-H) > **3c**(-Cl) > **3d**(-Br).

The IC_50_ values of tested compounds were also evaluated for CCD-18Co cells. As shown in Table 4, the IC_50_ values were in range 8–17 μM (**3b**–**3d**), and IC_50_ > 75 μM was for derivative **3a**. The effect of tested derivatives on the metabolic activity of normal fibroblasts evaluated by MTT assay decreased as follows **3a** < **3d** < **3c** < **3b**. The compounds with a halogen-substituted benzene ring (**3b**–**3d**) are more active in comparison to compounds without a halogen-substituted benzene ring (**3a**). However, doxorubicin was significantly the most efficient (IC_50_ = 5 μM) in affecting the metabolic activity of both cell lines. These results indicate that **3a** and **3b**–**3d** have > 15, 3 and 2 times less activity than doxorubicin in CCD18-Co normal colon fibroblasts, in comparison to the study of de Almedia et al. [73], where they synthetized 9-substitued acridine derivatives and tested their effect on metabolic activity (in concentration 0.25–250 mg/mL) against nine cancer lines. The non-halogenated compound decreased metabolic activity in each of the nine cell lines as compared to chlorinated (chloro) derivative, which was active against six cancer cell lines, and their effect on metabolic activity was more than seven times lower than the non-halogenated compounds. The brominated (bromo) compound decreased metabolic activity in only two cell lines. This effect on metabolic activity is similar to our observation support also our result in case the **3a**–**3d** where chlorinated (**3c**) and brominated (**3d**) compounds decreased metabolic activity less than the compound without a halogen-substituted benzene ring (**3a**).

The selectivity index (SI) for the effectiveness of the compound demonstrates its differential activity, where a higher level represents a higher selectivity [74]. An SI value less than 2 indicates general toxicity of the compound [75]. The SI values of **3a**–**3c** compounds for A549 cell line were determined and are present in Table 4. The values of SI indicate that all compounds exhibit a low selectivity (less than 2), and thus a general strong effect on cancer cell metabolism, except for derivative **3a**.

#### 2.4.2. The Effect of **3a** and **3c** on Viability, Cellularity, Clonogenic Survival, and Distribution of the Cell Cycle in the A549 Cancer Cell Line

A cell viability assay showed that compounds **3a** and **3c** in concentration 50 μM after 48 h incubation significantly (*p* < 0.001) decreased viability of A549 cells by 34% and 14%, as compared with the control group (Figure 9A). The 0.5% DMSO did not show a significant reduction in cell viability. The results from the cellularity assay showed that tested compounds induced a significant (*p* < 0.001) reduction of total cell number at both concentrations as compared with the control group (Figure 9B). For compounds **3a** and **3c** in a concentration of 25 μM, the number of cells was reduced similarly by 48%, 50% and 54%, respectively. In a higher concentration of 50 μM, the effect of acridine-benzohydrazides **3a** and **3c** was stronger (approximately 94% and 91% of reduction, respectively) than the control group. DMSO 0.5% showed no significant decrease in cell number. The derivatives **3a** and **3c** at lower concentration (25 μM) act as cytostatic agents rather than cytotoxic, but with increasing concentration (50 μM), their effect is simultaneously cytostatic and cytotoxic.

The MTT assay results provide information about the changes of metabolic activity linked to mitochondrial function, which does not always correlate with cell viability after treatment [18,72,74,75]. To assess drug efficacy and prevent treatment failure, it is important to observe whether cancer cells may proliferate during the interval between treatments. The clonogenic assay evaluates the efficacy of new drugs by observation of reduced repopulation of cancer cells and cell survival after treatment. This assay is important to select eligible anticancer drug candidates [76]. The clonogenic survival was analysed after treatment of A549 cells with the compounds **3a** and **3c**. DMSO (0.5%) was used as solvent control. A549 cells were treated with **3a** and **3c** in concentrations of 25 and 50 μM for 48 h. The cells were then counted, and 500 viable cells per well were seeded and cultivated for another seven days under standard conditions.

Figure 10A illustrates a plate of clonogenicity, where 500 vital cells A549 were seeded after 48 h treatment with **3a** and **3c** in two concentrations (25 and 50 μM) and 0.5% DMSO as a control. As shown in Figure 10B, acridine-benzohydrazides **3a** and **3c** in a concentration of 50 μM reduced the clonogenic ability of A549 cells by 72% or 74%, respectively, as compared to the control. However, treatment by compounds **3a** and **3c** in a concentration of 25 μM did not significantly reduce the ability of A549 cells to form new colonies. The obtained results are in good correlation with the results from previous viability and cellularity assays. The difference in substituted benzene ring in acridine-benzohydrazone (**3a**,**3c**) plays a slight role in clonogenicity survival (cellularity and viability, too) of A549 cells. However, halogenated compound **3c** (-Cl) is more efficient against CCD-18Co normal colon fibroblasts than non-halogenated compound **3a**.

The results from flow cytometry analysis of the cell cycle suggest that application of the compounds **3a** and **3c** primarily leads to accumulation of cells in the G2/M phase, followed by inhibition of A549 cells in G0/G1 phase, which subsequently result in inhibition of cell proliferation. G2/M phase is a critical phase before cell division [77].

Many widely used and potential cancer chemotherapeutic agents cause DNA damage by targeting DNA or enzymes that regulate DNA topology as topoisomerase I/II, resulting in DNA damage-induced G2/M arrest, which activates the apoptosis pathway (e.g., doxorubicin, amsacrine) [77,78,79,80]. However, not all potential anticancer drugs that arrest the cell cycle in G2/M phase were relevant to DNA damage, such as STK295900, DNA binding agents, and dual Topo I and Topo II catalytic inhibitors. Interestingly, STK295900 as a catalytic inhibitor of topoisomerase also acts as an antagonist on Topo poison-mediated DNA damage. Therefore, further study is needed to determine the mechanism underlying the **3a** and **3c** induced accumulation of A549 cells in the G2/M phase (Figure 11).

### 2.5. Inhibition of Topoisomerase I and II

The topoisomerase (Topo) inhibitors are molecules which (a) disrupt enzyme activity by forming ternary complex (DNA-Topo-compound)—these compounds were named as topoisomerase poisons—or (b) molecules which inhibit the catalytic function of enzymes—named catalytic inhibitors—and affect both to result in cell death (apoptosis) [73,81,82,83,84]. We performed Topo-mediated DNA relaxation or decatenation assays to test whether the new class of acridine derivatives **3a**–**3d** exert their antiproliferative function by targeting Topo.

Topo I enzymes convert the supercoiled (SC) form DNA to relaxed (R) form and nicked-open-circular (NOC) DNA. The NOC form is circular *ds*DNA with one nicked strength; Topo I nicked only one strength. SC form DNA migrate the fastest in gel, NOC migrates the slowest, and the migration speed of R DNA is in the middle of these two forms. In an electrophoretogram with active Topo I enzymes, SC DNA is not observed, but R and NOC DNA forms can be seen. In the case of Topo I inhibition, bends for R and NOC DNA forms are not observed [85,86]. The results electrophoretogram effect of derivatives **3a**–**3d** on activity topoisomerase I show Figure 12A. Interestingly, in the presence of increasing concentration of **3c** and **3d** derivatives, a decrease in R form and an increase in the SC form of DNA was observed, suggesting that these derivatives might inhibit the relaxation activity of Topo I. Therefore, these compounds in higher concentrations may act as catalytic Topo inhibitors [85].

The Topo IIα enzyme catalysed the decatenation of catenated *k*DNA to closed circular decatenated *k*DNA. In one specific case, linear (L) and nicked-opened (NC) forms of DNA could by observed. Figure 12 shows the result of the Topo Iiα decatenation assay in the presence of **3a**–**3d** derivatives in concentrations of 25, 50 and 100 μM. The *m*-Amsacrine (AMSA) (lane 2) was the observed band for catenated DNA. No obvious inhibitory effects were observed for compounds **3a** (lane 5–7) and **3b** (lane 8–10). However, derivatives **3c** (lane 12 and 13) and **3d** (lane 16) showed partial inhibition of Topo Iiα.

In previous studies, a direct correlation between the values of DNA binding constant of derivatives (for typically intercalating compounds) and their inhibition effect to Topo and antiproliferative activity (decreased metabolic activity of cancer cells) was observed [83,85,86]. However, the values of DNA binding constants were not a reliable guarantee of the inhibition of Topo by DNA binders’ derivatives. In a study of similar acridine derivatives with good binding constant values, only one compound was capable of inhibiting Topo I [18]. In our study, we observed a negative correlation between values of the DNA binding constant and the inhibition ability to *h*Topo I/IIα. Derivatives **3a** and **3b** with higher values of *K* did not have the potential to inhibit TopoI/IIα, thus derivatives **3c** and **3d** with lower values of *K* had the potential to inhibit TopoI/IIα. The derivatives **3c** and **3d** in vitro act as potential dual inhibitors of *h*Topo I and II with a partial effect on the metabolic activity of cancer cells A594. No direct correlation was observed between values of IC_50_ from the MTT assay (A549 cells) with the in vitro Topo inhibition assay. However, a positive correlation exists between values of IC_50_ and DNA binding constant (*K_b_*).

## 3. Materials and Methods

### 3.1. Materials and Physical Measurements

All the reagents were purchased from local suppliers and used without purification. The progress of the reaction was monitored using thin-layer chromatography (TLC). Analytical TLC was performed on pre-coated aluminium sheets of silica gel 60 F254 (Merck, Darmstadt, Germany), and the compounds were visualized using UV light. The melting points were determined on a Boetius apparatus. A stock solution of HSA was prepared in a NaCl-Tris-HCl (100 mM NaCl, 10 mM Tris) buffer solution (pH = 7.4) in distilled water. Tris(hydroxymetyl)aminomethane (Tris), HSA and NaCl were purchased from Sigma (St. Louis, MO, USA). The concentration of HSA in the stock solution was determined through absorption at 280 nm with a molar extinction coefficient ε_280_ = 35,700 M^−1^·cm^−1^ using spectrophotometric measurements. The solvents, chemicals and calf thymus DNA (ctDNA) used in this study were purchased from Sigma Aldrich and Lachema and used without further purification. Compounds **3a**–**3d** were dissolved in DMSO to produce a stock solution of 30 mM, with a working solution of 10 mM being prepared from this solution for further use. The solution of the compound was stored in dark conditions at −21 °C. The ctDNA was dissolved in a Tris-HCl-EDTA (pH = 8.3) (10 mM Tris-HCl (pH = 8.0); 1 mM EDTA) buffer by incubation at 4 °C with gentle mixing to form a homogenous solution over 24 h. The final concentration of the stock ctDNA solution was measured with a spectrophotometer using UV-Vis absorbance with a molar extinction coefficient of 6600 M^−1^ cm^−1^ at 260 nm. The purity of the ctDNA solution was determined by observing the ratio of the absorbance at 260 nm and 280 nm. The obtained absorbance ratio of A_260_/A_280_ = 1.84 indicates that the DNA was free from protein and had acceptable levels of purity for experimental use. The solution was stored at 4 °C for future use.

#### 3.1.1. NMR Spectra

Nuclear magnetic resonance data were collected on a Varian VNMRS 600 spectrometer operating at 599.87 MHz for ^1^H, 150.84 MHz for ^13^C, and 60.79 MHz for ^15^N. Chemical shifts (δ in ppm) are given from the internal solvent and the partially deuterated residual DMSO-d_6_ 39.5 ppm and acetone-d_6_ 29.8 ppm for ^13^C; DMSO-d_5_ 2.5 ppm and acetone-d_5_ 2.05 ppm for ^1^H. External nitromethane (0.0 ppm) was used for ^15^N references. The ^15^N chemical shifts were obtained from two dimensional ^1^H,^15^N-HMBC experiments with gradient coherence selection, which were performed using a standard pulse sequence from the Varian pulse library. CH_3_NO_2_ was used as an external reference for the ^15^N chemical shifts. The 2D experiments gCOSY, zTOCSY, NOESY, gHSQC and gHMBC were run using the standard Varian software. All data were analysed using MestReNova 14.2.1-27684 (5 May 2021, Santiago de Compostela, Spain) software.

#### 3.1.2. IR Spectra

The infrared spectra of the prepared compounds were recorded with an Avatar FT-IR 6700 (Fourier transform infrared spectroscopy) spectrometer at the wavenumber range of 4000–400 cm^−1^, with 64 repetitions for each spectrum using the ATR (attenuated total reflectance) technique. Prior to the measurements, samples were pressed with a rotary press to ensure sufficient contact with the surface of the diamond holder. All obtained data were analysed using Omnic 8.2.0.387 (2010) software.

#### 3.1.3. HR Mass Spectroscopy

The method used for high-resolution mass spectrometric identification of products is described in detail in the literature [87]. The following minor modifications were made to the published method: the samples were dissolved in chloroform (1 mg.mL^−1^) and diluted 1000-fold. An atmospheric solid analysis probe (ASAP) was dipped into the sample solution, placed into the ion source and analysed in full scan mode. The probe was kept at a constant temperature of 450 °C for 2 min. Mass accuracy of 1 ppm or less was achieved with the used instrumentation for all compounds.

### 3.2. Synthesis of Compounds ***3a***–***e***, ***4***, ***5*** and ***7b***–***d***

#### 3.2.1. Synthesis of *N*′-[(*E*)-acridin-4-yl)methylidene]benzohydrazides **3a**–**e**

Benzohydrazide (**1a**–**e**, 0.241 mmol) was added to a stirred suspension of acridine-4-carbaldehyde (**2**, 50 mg, 0.24 mmol) in dry ethanol (2 mL). The reaction mixture was refluxed until the acridine-9-carbaldehyde solution (**2**, TLC: dichloromethane/ethyl acetate, 4:1, *v*/*v*) was fully consumed. The reaction mixture was cooled, and the precipitate was filtered off and washed with dry ethanol. The crude product was crystallized from ethanol to give benzohydrazide **3**.

*N′-[(E)-(Acridin-4-yl)methylidene]benzohydrazide* (**3a**). Yield 71 mg, 90%; bright yellow solid; mp 235–236 °C; ^1^H NMR (600 MHz, DMSO-d_6_) *δ* 12.25 (1H, s, H-3), 9.93 (1H, s, H-1), 9.22 (1H, s, H-9′), 8.51 (1H, d, *J* 7.3 Hz, H-3′), 8.30 (1H, d, *J* 8.4 Hz, H-1′), 8.23 (1H, d, *J* 8.3 Hz, H-8′), 8.19 (1H, d, *J* 8.7 Hz, H-5′), 8.04 (2H, d, *J* 7.1 Hz, H-2″,6″), 7.93 (1H, ddd, *J* 8.8, 6.5, 1.4 Hz, H-6′), 7.76 (1H, t, *J* 7.2 Hz, H-2′), 7.68 (1H, ddd, *J* 8.0, 6.6, 1.1 Hz, H-7′), 7.63 (1 H, t, *J* 7.3 Hz, H-4″), 7.56 (2H, t, *J* 7.5 Hz, H-3″,5″). ^13^C NMR (151 MHz, DMSO-d_6_) *δ* 163.1 (C-4), 147.9 (C-10′a), 146.2 (C-4′a), 144.7 (C-1), 137.1 (C-9′), 133.4 (C-1″), 131.8 (C-4″), 131.2 (C-6′), 131.0 (C-4′), 130.5 (C-1′), 128.8 (C-5′), 128.7 (C-8′), 128.4 (C-3″), 127.8 (C-2″), 126.8 (C-3′), 126.2 (C-9′a), 126.2 (C-8′a), 126.2 (C-7′), 125.7 (C-2′). ^15^N NMR (61 MHz, DMSO-d_6_) *δ* −206.0 (N-3), −83.3 (N-10′), −57.3 (N-2).

*N′-[(E)-Acridin-4-ylmethylidene]-4-fluorobenzohydrazide* (**3b**). Yield 68 mg, 82%; bright yellow solid; mp 229–230 °C; ^1^H NMR (600 MHz, DMSO-d_6_) *δ* 12.26 (1H, s, H-3), 9.91 (1H, s, H-1), 9.23 (1H, s, H-9′), 8.51 (1H, d, *J* 6.7 Hz, H-3′), 8.31 (1H, d, *J* 8.6 Hz, H-1′), 8.23 (1H, d, *J* 8.7 Hz, H-8′), 8.19 (1H, d, *J* 8.5 Hz, H-5′), 8.11 (2H, d, *J* 8.4, 5.4 Hz, H-2″,6″), 7.93 (1H, ddd, *J* 8.5, 6.6, 1.4 Hz, H-6′), 7.76 (1H, t, *J* 7.7 Hz, H-2′), 7.68 (1H, ddd, *J* 8.1, 6.5, 1.0 Hz, H-7′), 7.40 (2H, d, *J* 8.7 Hz, H-3″,5″); ^13^C NMR (151 MHz, DMSO-d_6_) *δ* 164.2 (d, *J* 248 Hz, C-4″), 162.1 (C-4), 147.9 (C-10′a), 146.2 (C-4′a), 144.8 (C-1), 137.1 (C-9′), 131.3 (C-6′), 130.9 (C-4′), 130.6 (C-1′), 130.5 (d, *J* 9 Hz, C-2″), 129.8 (C-1″), 128.8 (C-5′), 128.7 (C-8′), 126.8 (C-3′), 126.2 (C-9′a), 126.2 (C-8′a), 126.2 (C-7′), 125.7 (C-2′), 128.6 (d, *J* 21.6 Hz, C-3″); ^15^N NMR (61 MHz, DMSO-d_6_) *δ* −206.3 (N-3), −57.6 (N-2).

*N′-[(E)-Acridin-4-ylmethylidene]-4-chlorobenzohydrazide* (**3c**). Yield 79 mg, 91%; bright yellow solid; mp 266–267 °C; ^1^H NMR (600 MHz, DMSO-d_6_) *δ* 12.30 (1H, s, H-3), 9.92 (1H, s, H-1), 9.23 (1H, s, H-9′), 8.51 (1H, d, *J* 5.9 Hz, H-3′), 8.31 (1H, d, *J* 7.6 Hz, H-1′), 8.23 (1H, d, *J* 8.6 Hz, H-8′), 8.19 (1H, d, *J* 9.3 Hz, H-5′), 8.06 (2H, d, *J* 8.5 Hz, H-2″,6″), 7.93 (1H, ddd, *J* 8.5, 6.5, 1.4 Hz, H-6′), 7.76 (1H, t, *J* 7.7 Hz, H-2′), 7.68 (1H, ddd, *J* 8.1, 6.6, 1.1 Hz, H-7′), 7.64 (2H, d, *J* 8.5 Hz, H-3″,5″); ^13^C NMR (151 MHz, DMSO-d_6_) *δ* 162.1 (C-4), 147.9 (C-10′a), 146.2 (C-4′a), 145.0 (C-1), 137.1 (C-9′), 136.6 (C-4″), 132.1 (C-1″), 131.3 (C-6′), 130.9 (C-4′), 130.6 (C-1′), 129.7 (C-2″), 128.8 (C-5′), 128.7 (C-8′), 128.6 (C-3″), 126.8 (C-3′), 126.2 (C-9′a), 126.2 (C-8′a), 126.2 (C-7′), 125.7 (C-2′). ^15^N NMR (61 MHz, DMSO-d_6_) *δ* −206.0 (N-3), −57.9 (N-2).

*N′-[(E)-Acridin-4-ylmethylidene]-4-bromobenzohydrazide* (**3d**). Yield 71 mg, 73%; bright yellow solid; mp 269–270 °C; ^1^H NMR (600 MHz, DMSO-d_6_) *δ* 12.30 (1H, s, H-3), 9.92 (1H, s, H-1), 9.22 (1H, s, H-9′), 8.50 (1H, d, *J* 6.3 Hz, H-3′), 8.30 (1H, d, *J* 9.2 Hz, H-1′), 8.23 (1H, d, *J* 8.5 Hz, H-8′), 8.19 (1H, d, *J* 8.8 Hz, H-5′), 7.99 (2H, d, *J* 8.4 Hz, H-2″,6″), 7.93 (1H, ddd, *J* 8.5, 6.6, 1.4 Hz, H-6′), 7.78 (2H, d, *J* 8.5 Hz, H-3″,5″), 7.75 (1H, t, *J* 7.2 Hz, H-2′), 7.67 (1H, t, *J* 7.99 Hz, H-7′); ^13^C NMR (151 MHz, DMSO-d_6_) *δ* 162.2 (C-4), 147.9 (C-10′a), 146.2 (C-4′a), 145.1 (C-1), 137.1 (C-9′), 132.4 (C-1″), 131.5 (C-3″), 131.3 (C-6′), 130.9 (C-4′), 130.6 (C-1′), 129.9 (C-2″), 128.8 (C-5′), 128.7 (C-8′), 126.8 (C-3′), 126.2 (C-9′a), 126.2 (C-8′a), 126.2 (C-7′), 125.7 (C-2′), 125.6 (C-4″); ^15^N NMR (61 MHz, DMSO-d_6_) *δ* −206.0 (N-3), −58.0 (N-2).

*N′-[(E)-Acridin-4-ylmetylidene]-4-methoxybenzohydrazide* (**3e**). Yield 70 mg, 82%; bright yellow solid; mp 249–250 °C; ^1^H NMR (600 MHz, DMSO-d_6_) *δ* 12.12 (1H, s, H-3), 9.90 (1H, s, H-1), 9.22 (1H, s, H-9′), 8.49 (1H, d, *J* 6.5 Hz, H-3′), 8.29 (1H, d, *J* 8.3 Hz, H-1′), 8.23 (1H, d, *J* 8.3 Hz, H-8′), 8.20 (1H, d, *J* 8.7 Hz, H-5′), 8.03 (2H, d, *J* 8.4 Hz, H-2″,6″), 7.93 (1H, ddd, *J* 8.4, 6.5, 1.4 Hz, H-6′), 7.75 (1H, t, *J* 7.7 Hz, H-2′), 7.68 (1H, t, *J* 7.5 Hz, H-7′), 7.09 (2H, d, *J* 8.8 Hz, H-3″,5″), 3.86 (3H, s, CH_3_O); ^13^C NMR (151 MHz, DMSO-d_6_) *δ* 162.5 (C-4), 162.0 (C-4″), 147.9 (C-10′a), 146.2 (C-4′a), 144.0 (C-1), 137.1 (C-9′), 131.2 (C-6′), 131.1 (C-4′), 130.4 (C-1′), 129.7 (C-2″), 128.8 (C-5′), 128.7 (C-8′), 126.6 (C-3′), 126.2 (C-9′a), 126.2 (C-8′a), 126.2 (C-7′), 125.7 (C-2′), 125.4 (C-1″), 113.7 (C-3″), 55.4 (CH_3_O); ^15^N NMR (61 MHz, DMSO-d_6_) *δ* −207.3 (N-3), −83.3 (N-10′), −56.6 (N-2).

#### 3.2.2. Synthesis of Methyl Acridine-4-Carboxylate (**4**)

Methanolic solutions (each 8 mL) of iodine (1.176 g, 4.63 mmol) and KOH (0.455 g, 8.11 mmol) were added alternately at 0 °C to a solution of aldehyde 2 (600 mg, 2.90 mmol) in absolute methanol (8 mL) cooled to 0 °C. The reaction mixture was stirred for 2 h at 0 °C, and then for 4 h at room temperature. The reaction mixture was poured into an aqueous solution of Na_2_S_2_O_3_ × 5 H_2_O (1 M, 50 mL). The resulting yellow precipitate was filtered off and washed with cold water (2–10 mL). The crude product **4** was crystallized from methanol.

#### 3.2.3. Synthesis of Acridine-4-Carbohydrazide (**5**)

Hydrazine monohydrate (50–60%, 0.69 mL, 22.25 mmol) was added to a stirred suspension of methyl acridine-4-carboxylate (**4**, 330 mg, 1.39 mmol) in ethanol (3 mL). The reaction mixture was refluxed for 3 h. Subsequently, a further quantity of hydrazine monohydrate (50–60%, 0.69 mL, 22.25 mmol) was added. The reaction mixture was refluxed until the methyl acridine-4-carboxylate (**4**, TLC: dichloromethane/ethyl acetate, 9:1 *v*/*v*) was fully consumed. The reaction mixture was cooled, and the precipitate was filtered off and washed with cold absolute ethanol. The filtrate was concentrated to produce another portion of product **5**. The crude product **5** was crystallized from ethanol.

Yield 515 mg, 75%; brown solid; mp 225–226 °C; ^1^H NMR (600 MHz, DMSO-d_6_) *δ* 12.12 (1H, s, H-2), 9.36 (1H, s, H-9′), 8.72 (1H, dd, *J* = 7.0, 1.6 Hz, H-3′), 8.39 (1H, ddd, *J* = 8.4, 1.6, 0.6 Hz, H-1′), 8.30 (1H, dd, *J* = 8.7, 0.9 Hz, H-5′), 8.26 (1H, dt, *J* = 8.4, 0.7 Hz, H-8′), 7.99 (1H, ddd, *J* = 8.7, 6.6, 1.4 Hz, H-6′), 7.77 (1H, dd, *J* = 8.3, 7.0 Hz, H-2′), 7.73 (1H, ddd, *J* = 8.4, 6.6, 1.2 Hz, H-7′), 4.91 (2H, d, *J* = 4.4 Hz, H-3). ^13^C NMR (151 MHz, DMSO-d_6_) *δ* 164.0 (C-1), 147.2 (C-10′a), 145.2 (C-4′a), 138.8 (C-9′), 134.2 (C-3′), 132.7 (C-1′), 132.1 (C-6′), 128.6 (C-8′), 128.5 (C-5′), 127.9 (C-4′), 126.6 (C-7′), 126.5 (C-9′a), 125.8 (C-4′a), 125.3 (C-2′).

#### 3.2.4. Synthesis of *N*′-[(*E*)-Phenylmethylidene]acridine-4-carbohydrazides **7b**–**d**

Benzaldehyde **6** (0.422 mmol) was added to a stirred suspension of acridine-4-carbohydrazide (**5**, 100 mg, 0.42 mmol) in ethanol (2 mL). The reaction mixture was refluxed until the benzaldehyde **6** (TLC: dichloromethane/ethyl acetate, 4:1 *v*/*v*) was fully consumed. The reaction mixture was cooled, and the precipitate was filtered off and washed with cold absolute ethanol. The crude product was crystallized from methanol to give carbohydrazide **7**.

*N′-[(E)-4-Fluorophenylmethylidene]acridine-4-carbohydrazide* (**7b**). Yield 130 mg, 90%; bright yellow solid; mp 255–256 °C. Ratio of isomers: *Z*_N-N_: *E*_N-N_ = 1.0: 0.65 (60.6%: 39.4%). Isomer *Z*_N-N_: ^1^H NMR (600 MHz, CD_3_COCD_3_) *δ* 14.98 (1H, s, H-2), 9.32 (1H, s, H-9′), 9.00 (1H, d, *J* 7.0 Hz, H-3′), 8.88 (1H, s, H-4), 8.49 (1H, d, *J* 8.8 Hz, H-5′), 8.44 (1H, dd, *J* 8.3, 1.6 Hz, H-1′), 8.27 (1H, d, *J* 8.5 Hz, H-8′), 8.02 (1H, m, H-6′), 7.99 (2H, dd, *J* 8.8, 5.6 Hz, H-2″,6″), 7.82 (1H, dd, *J* 8.3, 7.1 Hz, H-2′), 7.75 (1H, ddd, *J* 8.0, 6.6, 1.0 Hz, H-7′), 7.27 (2H, t, *J* 8.7 Hz, H-3″,5″); ^13^C NMR (151 MHz, CD_3_COCD_3_) *δ* 164.68 (C-4″), 162.62 (C-1), 148.54 (C-10′a), 147.73 (C-4), 146.68 (C-4′a), 139.95 (C-9′), 136.81 (C-3′), 134.40 (C-1′), 132.94 (C-6′), 132.74 (C-1″), 130.34 (C-2″,6″), 129.63 (C-5′), 129.45 (C-8′), 128.92 (C-4′), 127.98 (C-9′a), 127.61 (C-7′), 127.23 (C-8′a), 126.36 (C-2′), 116.54 (C-3″,5″); ^15^N NMR (61 MHz, CD_3_COCD_3_) *δ* −200.56 (N-2), −95.88 (N-10′), −60.19 (N-3). Isomer *E*_N-N_: ^1^H NMR (600 MHz, CD_3_COCD_3_): *δ* 14.98 (1H, s, H-2), 9.32 (1H, s, H-9′), 9.00 (1H, d, *J* 7.0 Hz, H-3′), 8.86 (1H, s, H-4), 8.49 (1H, d, *J* 8.8 Hz, H-5′), 8.44 (1H, dd, *J* 8.3, 1.6 Hz, H-1′), 8.27 (1H, d, *J* 8.5 Hz, H-8′), 8.02 (1H, m, H-6′), 7.99 (2H, dd, *J* 8.8, 5.6 Hz, H-2″,6″), 7.82 (1H, dd, *J* 8.3, 7.1 Hz, H-2′), 7.75 (1H, ddd, *J* 8.0, 6.6, 1.0 Hz, H-7′), 7.27 (2H, t, *J* 8.7 Hz, H-3″,5″); ^13^C NMR (151 MHz, CD_3_COCD_3_) *δ* 164.68 (C-4″), 162.52 (C-1), 148.50 (C-10′a), 147.71 (C-4), 146.68 (C-4′a), 139.98 (C-9′), 136.81 (C-3′), 134.37 (C-1′), 132.96 (C-6′), 132.74 (C-1″), 130.34 (C-2″,6″), 129.59 (C-5′), 129.45 (C-8′), 128.92 (C-4′), 127.98 (C-9′a), 127.61 (C-7′), 127.23 (C-8′a), 126.37 (C-2′), 116.54 (C-3″,5″); ^15^N NMR (61 MHz, CD_3_COCD_3_) *δ* −199.45 (N-2), −95.88 (N-10′), −59.69 (N-3).

*N′-[(E)-4-Chlorophenylmethylidene]acridine-4-carbohydrazide* (**7c**). Yield 121 mg, 80%; bright yellow solid; mp 243–244 °C. Ratio of isomers *Z*_N-N_: *E*_N-N_ = 1.0: 1.21 (45.2%: 54.8%). Isomer *Z*_N-N_: ^1^H NMR (600 MHz, CD_3_COCD_3_) *δ* 15.00 (1H, s, H-2), 9.28 (1H, s, H-9′), 8.98 (1H, d, *J* 7.0 Hz, H-3′), 8.85 (1H, s, H-4), 8.46 (1H, d, *J* 8.8 Hz, H-5′), 8.41 (1H, dd, *J* 8.3, 1.6 Hz, H-1′), 8.24 (1H, d, *J* 8.5 Hz, H-8′), 7.99 (1H, ddd, *J* 8.4, 6.6, 1.2 Hz, H-6′), 7.94 (2H, d, *J* 8.5 Hz, H-2″,6″), 7.80 (1H, dd, *J* 8.3, 7.1 Hz, H-2′), 7.73 (1H, ddd, *J* 8.4, 6.6, 1.2 Hz, H-7′), 7.52 (2H, d, *J* 8.5 Hz, H-3″,5″); ^13^C NMR (151 MHz, CD_3_COCD_3_) *δ* 162.59 (C-1), 148.49 (C-10′a), 147.56 (C-4), 146.61 (C-4′a), 139.96 (C-9′), 136.83 (C-3′), 136.01 (C-4″), 135.09 (C-1″), 134.45 (C-1′), 132.93 (C-6′), 129.75 (C-2″,6″,C-3″,5″), 129.56 (C-5′), 129.43 (C-8′), 128.78 (C-4′), 127.93 (C-9′a), 127.59 (C-7′), 127.20 (C-8′a), 126.32 (C-2′); ^15^N NMR (61 MHz, CD_3_COCD_3_) *δ* −199.98 (N-2), −96.30 (N-10′), −57.86 (N-3). Isomer *E*_N-N_: ^1^H NMR (600 MHz, CD_3_COCD_3_) *δ* 15.00 (1H, s, H-2), 9.28 (1H, s, H-9′), 8.98 (1H, d, *J* 7.0 Hz, H-3′), 8.83 (1H, s, H-4), 8.46 (1H, d, *J* 8.8 Hz, H-5′), 8.41 (1H, dd, *J* 8.3, 1.6 Hz, H-1′), 8.24 (1H, d, *J* 8.5 Hz, H-8′), 7.99 (1H, ddd, *J* 8.4, 6.6, 1.2 Hz, H-6′), 7.94 (2H, d, *J* 8.5 Hz, H-2″,6″), 7.80 (1H, dd, *J* 8.3, 7.1 Hz, H-2′), 7.73 (1H, ddd, *J* 8.4, 6.6, 1.2 Hz, H-7′), 7.52 (2H, d, *J* 8.5 Hz, H-3″,5″); ^13^C NMR (151 MHz, CD_3_COCD_3_) *δ* 162.69 (C-1), 148.44 (C-10′a), 147.54 (C-4), 146.61 (C-4′a), 139.92 (C-9′), 136.83 (C-3′), 136.00 (C-4″), 135.09 (C-1″), 134.43 (C-1′), 132.95 (C-6′), 129.75 (C-2″,6″,C-3″,5″), 129.60 (C-5′), 129.43 (C-8′), 128.78 (C-4′), 127.93 (C-9′a), 127.59 (C-7′), 127.20 (C-8′a), 126.34 (C-2′); ^15^N NMR (61 MHz, CD_3_COCD_3_) *δ* −198.95 (N-2), −96.30 (N-10′), −57.34 (N-3).

*N′-[(E)-4-Bromophenylmethylidene]acridine-4-carbohydrazide* (**7d**). Yield 140 mg, 82%; bright yellow solid; mp 240–241 °C. Ratio of isomers *Z*_N-N_: *E*_N-N_ = 1.0: 0.48 (67.6%: 32.4%). Isomer *Z*_N-N_***:***
^1^H NMR (600 MHz, CD_3_COCD_3_) *δ* 15.05 (1H, s, H-2), 9.32 (1H, s, H-9′), 9.00 (1H, d, *J* 7.0 Hz, H-3′), 8.85 (1H, s, H-4), 8.49 (1H, d, *J* 8.5 Hz, H-5′), 8.44 (1H, dd, *J* 8.3, 1.6 Hz, H-1′), 8.26 (1H, d, *J* 8.5 Hz, H-8′), 8.01 (1H, ddd, J 8.4, 6.6, 1.2 Hz, H-6′), 7.88 (2H, d, *J* 8.5 Hz, H-2″,6″), 7.82 (1H, dd, *J* 8.3, 7.1 Hz, H-2′), 7.74 (1H, ddd, *J* 8.4, 6.6, 1.2 Hz, H-7′), 7.69 (2H, d, *J* 8.5 Hz, H-3″,5″); ^13^C NMR (151 MHz, CD_3_COCD_3_) *δ* 162.54 (C-1), 148.48 (C-10′a), 147.58 (C-4), 146.61 (C-4′a), 139.99 (C-9′), 136.85 (C-3′), 135.46 (C-1″), 134.47 (C-1′), 132.94 (C-6′), 132.73 (C-3″,5″), 129.95 (C-2″,6″), 129.60 (C-5′), 129.43 (C-8′), 128.75 (C-4′), 127.93 (C-9′a), 127.60 (C-7′), 127.19 (C-8′a), 126.34 (C-2′), 124.32 (C-4″); ^15^N NMR (61 MHz, CD_3_COCD_3_) *δ* −199.57 (N-2), −96.30 (N-10′), −57.54 (N-3). Isomer *E*_N-N_: ^1^H NMR (600 MHz, CD_3_COCD_3_) *δ* 15.05 (1H, s, H-2), 9.32 (1H, s, H-9′), 9.00 (1H, d, *J* 7.0 Hz, H-3′), 8.84 (1H, s, H-4), 8.49 (1H, d, *J* 8.5 Hz, H-5′), 8.44 (1H, dd, *J* 8.3, 1.6 Hz, H-1′), 8.26 (1H, d, *J* 8.5 Hz, H-8′), 8.01 (1H, ddd, J 8.4, 6.6, 1.2 Hz, H-6′), 7.88 (2H, d, *J* 8.5 Hz, H-2″,6″), 7.82 (1H, dd, *J* 8.3, 7.1 Hz, H-2′), 7.74 (1H, ddd, *J* 8.4, 6.6, 1.2 Hz, H-7′), 7.69 (2H, d, *J* 8.5 Hz, H-3″,5″); ^13^C NMR (151 MHz, CD_3_COCD_3_) *δ* 162.64 (C-1), 148.44 (C-10′a), 147.56 (C-4), 146.61 (C-4′a), 139.96 (C-9′), 136.85 (C-3′), 135.46 (C-1″), 134.45 (C-1′), 132.96 (C-6′), 132.73 (C-3″,5″), 129.95 (C-2″,6″), 129.56 (C-5′), 129.43 (C-8′), 128.75 (C-4′), 127.93 (C-9′a), 127.60 (C-7′), 127.19 (C-8′a), 126.35 (C-2′), 124.31 (C-4″); ^15^N NMR (61 MHz, CD_3_COCD_3_) *δ* −198.88 (N-2), −96.30 (N-10′), −57.17 (N-3).

### 3.3. HSA Measurement

#### 3.3.1. Steady-State Fluorescence, Synchronous and 3D Spectra

Steady-state fluorescence (STF) emission spectra, synchronous fluorescence (SF) spectra, and 3D fluorescence spectra (3DF) were measured using a Varian Cary Eclipse spectrofluorimeter with a xenon flash lamp and a 1.0 cm quartz cuvette, with the slits set at 5 nm for the excitation observations and 10 nm for the emission spectra. The STF spectra were recorded at a range of 285–550 nm with a fixed excitation wavelength at 280 nm. The change in fluorescence intensity of HSA at a concentration of 4 μM was observed by titrating varying concentrations of acridine derivatives **3a**–**3d** (0–6.2 μM) at three different temperatures (25, 30 and 35 °C) in 2 mL of 100 mM NaCl and a 10 mM Tris-HCl buffer (pH = 7.4).

Synchronous fluorescence spectra for HSA (4 μM) were recorded at increasing concentrations of compounds **3a**–**3d** in the same concentration range as that used in the STF studies. The spectra were recorded at a range of 200–400 nm by setting ∆λ = 60 nm and ∆λ = 15 nm for tryptophan and tyrosine residues, respectively, at room temperature (25 °C).

The 3DF spectra of HSA were performed in the absence and presence of compounds **3a**–**3d** using an excitation wavelength range of 200–350 nm and an emission wavelength range of 200–600 nm at room temperature (25 °C). The 3D spectra were recorded for 4 μM of HSA in a 2 mL buffer solution (100 mM NaCl and 10 mM Tris-HCl buffer pH = 7.4) and for the HSA:**3a**–**3d** complexes at a concentration ratio 1:1. The data from 3D measurements were processed into a 3D graphics plot using Origine 8.5 software, (2020) produced by OriginLab Corporation, Informer Technologies, Inc. (Los Angeles, CA, USA).

#### 3.3.2. Competitive Experiments

Competitive experiments of HSA were performed using warfarin and ibuprofen as standard site marker ligands. The selected markers for the site I were warfarin, whereas site II was probed using ibuprofen. HSA-acridine derivative (**3a**–**3d**) complexes at a concentration ratio of 1:1 ([HSA] = 4 μM) were titrated into specific side markers (0–20 μM). The molar ratio of the HSA:marker complexes were 1:0.5, 1:1, 1:1.5, 1:2, 1:2.5, 1:3, 1:3.5, 1:4, 1:4.5 and 1:5. The reaction mixture of the HSA-acridine derivatives complexes with the side markers were preincubated for 15 min prior to spectral measurements. Additionally, a reverse titration competitive experiment was performed for derivative **3a**. The site probe-HSA complex (1:1, [HSA] = 4 μM) was titrated with derivative **3a** at a concentration gradient of 0.6–6 μM. In order to determine the binding site of compound **3a**, the fluorescence quenching data were analysed using Stern-Volmer and modified Stern-Volmer equations to calculate the value of the Stern-Volmer constants (*K*_SV_) and binding constants *K_b_*, and to determine the number of binding sites n. All spectra measurements were performed at 280 nm excitation wavelength and the slits were set at 5 nm for excitation and 10 nm for emission spectra at a range of 290–550 nm and at 25 °C.

#### 3.3.3. ctDNA Binding Experiments

UV-Vis absorption spectrum of the drug-ctDNA complexes were measured on a Varian Cary 100 Bio UV-Vis Spectrophotometer. The UV-Vis spectra of free compounds **3a**–**3d** and drug-ctDNA complexes were recorded at the wavelength range of 220 to 600 nm. The measurements were performed in a 1.0 cm quartz cuvette with 2 mL of a Tris-HCl (10 mM, pH = 7.4) buffer at room temperature. The titration experiment was carried out in the presence of a fixed concentration of compounds **3a**–**3d** (25 μM) and was performed by titrating varying concentrations of ctDNA ranging from 0 to 680 μM. The solution was incubated for 5 min and then tested.

### 3.4. In Vitro Antiproliferative Assay

#### 3.4.1. Cell Culture and Treatment

Human lung carcinoma cell line A549 and CCD-18Co colon fibroblasts were purchased from the American Type Culture Collection (ATCC, Rockville, MD, USA). The A549 cells were cultured in a complete RPMI-1640 medium (Sigma-Aldrich, St. Louis, MO, USA) and CCD-18Co cells were cultured in a minimum essential medium (MEM) (PAN-Biotech GmbH, Aidenbach, Germany) at 37 °C, 95% humidity and 5% CO_2_. The media were supplemented with 10% fetal bovine serum (FBS, Biosera, Nuaille, France) and antibiotics (1% Antibiotic-Antimycotic 100 × and 50 × 10^−3^ g L^−1^ gentamicin, Biosera). Prior to the selected treatments, cells were seeded on 6- and/or 96-well plates (TPP, Trasadingen, Switzerland) and left to settle for 24 h. The acridine compounds solutions (at concentrations ranging from 5–75 µM) were then added to cells for 24 or 48 h, and analysis was subsequently performed.

#### 3.4.2. MTT Assay

MTT assays were performed in order to evaluate changes in the metabolic activity of cells that had occurred as a consequence of treatment with the acridine compounds. A549 cells (15 × 10^3^ cells/cm^2^) and CCD-18Co (15.625 × 10^3^ cells/cm^2^) were seeded in 96-well microplates. The A549 cells were treated for 24 and 48 h, and the CCD-18Co cells were treated for 48 h with different concentrations (5, 25, 50 and 75 µM) of the derivatives. After the treatment, the MTT (3-(4,5-dimethylthiazol-2-yl)-2,5-diphenyltertrazolium bromide) solution in PBS (5 mg mL^−1^) was added to each well. The reaction was stopped after 4 h incubation, and the formazan was dissolved by the addition of SDS at a final concentration of 3.3%. The absorbance of formazan (λ = 584 nm) was measured using a BMG FLUOstar Optima spectrometer (BMG Labtech GmbH, Offenburg, Germany). The results were evaluated as percentages of the absorbance of the untreated control. Results are presented as the average percentage of cells from three independent experiments. IC_50_ values for the derivatives were extrapolated from a sigmoidal fit (dose-response curve) to the metabolic activity data using OriginPro 8.5.0 SR1 (OriginLab Corp., Northampton, MA, USA).

#### 3.4.3. Quantification of Cell Number and Viability

For the assessment of total cell numbers and viability within individual experimental groups, floating and adherent cells were harvested after treatment with the studied compounds and evaluated using a Bürker chamber with eosin staining. A549 cells were plated at a density of 135 × 10^3^ cell/well into a 6-well plate and treated with different concentration of derivatives (25 and 50 µM) for 48 h. The total cell number was expressed as a percentage of the untreated control of the total cell number. Viability was expressed as a percentage of viable, eosin negative cells. The results are presented as the average percentage of cells from three independent experiments.

#### 3.4.4. Colony Forming Assay

For the colony forming assay, floating and adherent cells were harvested together 48 h after treatment with the studied compounds (25 and 50 µM). The cells were then counted using a Bürker chamber with eosin staining and 500 viable cells per well were seeded in 6-well plates. After seven days of cultivation under standard conditions, the cells in the plates were fixed and stained with 1% methylene blue dye in methanol. Visualized colonies were scanned and counted by Image software, and the results were evaluated as percentages of the untreated control. The results are presented as the average percentage of colonies from three independent experiments.

#### 3.4.5. Cell Cycle Analysis

For flow cytometric analysis of the cell cycle distribution, floating and adherent cells were harvested together 48 h after treatment with the compounds (25 and 50 µM), washed in cold PBS, fixed in cold 70% ethanol, and stored overnight at −20 °C. Prior to analysis, the cells were washed twice in PBS, resuspended in a staining solution (0.1% Triton X-100, 0.137 g L^−1^ ribonuclease A and 0.02 g L^−1^ propidium iodide (PI) and incubated in dark conditions at RT for 30 min. The DNA content was analysed using a BD FACSCalibur flow cytometer (Becton Dickinson, San Jose, CA, USA) with a 488 nm argon-ion excitation laser, and fluorescence was detected using a 585/42 nm band-pass filter (FL-2). ModFit 3.0 software (Verity Software House, Topsham, ME, USA) was used to generate DNA content frequency histograms and to quantify the percentage of cells in the individual cell cycle phases. The results are presented as the average ratio of cells in the individual phase to all cells from three independent experiments.

#### 3.4.6. Selectivity Index (SI)

SI = IC_50_ of pure compound in a normal cell line/IC_50_ of the same pure compound in a cancer cell line, where IC_50_ is the concentration that induced 50% inhibition on the growth of the treated cells.

### 3.5. Statistical Analysis

The obtained data were analysed using a one-way ANOVA with Tukey’s post-test, and are expressed as the mean ± standard deviation (S.D.) of at least three independent experiments. The experimental groups treated with derivatives were compared with the control group: (*): *p* ˂ 0.05, (**): *p* ˂ 0.01, (***): *p* ˂ 0.001.

## 4. Conclusions

A series of novel benzohydrazide derivatives **3a**–**3d** containing acridine moiety were designed, synthesized and characterized in detail using NMR, IR and HR mass spectroscopy techniques. NMR derived parameters (heteronuclear one and two bond coupling constants and ^1^H, ^13^C, and ^15^N chemical shifts) allowed us to determine configuration and conformation in solution for all synthesized compounds. We determined distinct configurational and conformational preferences to form *E*_C4=N3_*Z*_C(O)-N2_-**3** and *E*_C4=N3_*Z*_C(O)-N2_-**7**. The duplicate signals in the NMR spectra of **7b**–**d** were attributed to the present *E*_N-N_/*Z*_N-N_ conformers. The major factor that controls the conformation of the studied compound **7** are hydrogen bonds N10′···H2 and C(O)···H4.

In vitro antiproliferative activities of these compounds against A549 and normal fibroblast cells CCD-18Co were studied. The compounds have been undergoing against topoisomerase I and II, and their binding properties (ctDNA, HSA) have been evaluated. The derivatives **3c** and **3d** in vitro act as potential dual inhibitors of *h*Topo I and II with a partial effect on the metabolic activity of A594 cancer cells. The values of binding constant for the interaction of acridine derivatives with ctDNA increased as follows: **3d** < **3c** < **3a** < **3b**. The higher values of *K* from the acridine-benzohydrazone series are present in fluoro-substituted derivative **3b** (*K* = 3.18 × 10^3^ M^−1^). The effect of tested derivatives on the metabolic activity of A549 cells evaluated by MTT assay decreased as follows: **3b**(-F) > **3a**(-H) > **3c**(-Cl) > **3d**(-Br). In the case of **3d**, no significant activity against CCD-18Co fibroblasts was observed. The clonogenic survival was analysed after treatment of A549 cells with the compounds **3a** and **3c**. The acridine-benzohydrazides **3a** and **3c** reduced the clonogenic ability of A549 cells by 72% or 74%, respectively. The difference in the substituted benzene ring in **3a**, **3c** plays a slight role in clonogenicity survival (cellularity and viability, too) of A549 cell. The results indicated that interaction between acridine derivatives and HSA could lead to the change of protein secondary structure. In the presence of warfarin, the values of binding constants decreased, which suggest that derivative **3a** could bind to HSA at the Sudlow site I. The findings presented in this paper suggest that these acridine derivatives exhibit promising potential as topoisomerase I and II inhibitors with anticancer activity against A549 human adherent lung carcinoma cells, and may also serve as DNA and HSA-interacting agents. These features would be of considerable use in the development of drugs with enhanced or more selective effects and greater clinical efficacy.

## Data Availability

Data are contained within the article and Appendix A.

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
