# Peer review of "Acridine Based N-Acylhydrazone Derivatives as Potential Anticancer Agents: Synthesis, Characterization and ctDNA/HSA Spectroscopic Binding Properties"

_molecules, 2022, doi:10.3390/molecules27092883_

Round 1

Reviewer 1 Report

The manuscript by M. Vilkova et al. report on the synthesis of a series of acridine-derived acylhydrazones designed as potential anticancer agents. The authors prepared five compounds starting with acridin-4-carbaldehyde and selected benzohydrazides functionalized at para position with H, F, Cl, Br, and OMe group. carbohydrazide The second series of isomeric hydrazones (three compouds) was obtained by using the respective acridine-based carbohydrazide and p-halogenated benzaldehydes. The reported syntheses follow general protocols for functional group interconversions of carbonyl compounds as well as the condensations leading to acyl-hydrazones. The target products (and also some intermediates) are well characterized and the data provided in SI (NMR, IR, HRMS) confirm the expected structure. In addition, by means of NMR measurements configuration along the C=N bond (E-isomer only) was determined and the impact of H-bonding on conformational effects along the formal sigma N-N and C-N bonds in selected solvents (DMSO and acetone) was analyzed. The obtained materials were examined for antiproliferative activity against lung cancer A549 cell lines (and normal fibroblast CCD-18Co as a reference) to show moderate effectiveness in comparison to doxorubicin (DXR) and only low selectivity. Binding properties with ctDNA and HAS have been checked and promising potential inhibitory activity with topoisomerase I and II was indicated. Overall, the presented paper is of some importance and can be considered for publication, however, the Authors are encouraged to address the following points in preparation of the revised version.

- line 31: the general statement about the type of heteroatom-activity mechanism relationship should be supported with a series of appropriate general reviews rather that a single research article (ref.1)

- lines 49-52: this sentence should be supplemented with a proper citation.

- Introduction: the concept of combining of acridine pharmacophore with acylhydrazone unit for applications in biology and material sciences is known. Some recent work deserve mentioning (e.g. Aarjane et al. Arab. J Chem. 2020, 13, 6236; Ma et al. Soft. Matter 2021, 17, 7813)

- chapter 2.4.1.: the decreasing effect of metabolic activity of A549 follows the decreasing stability of sigma bonds in the tested compounds of series 3 (F > H > Cl > Br). This observation should be briefly discussed.

Further minor corrections:

Abstract, line 18: remove semicolon to read ‘complex 3b’

Abstract, line 23: compound 3a should be written bold

p.4, Figure 4: for the sake of the readers convenience, the depicted structures should be larger. Also, some hydrogen atoms in these structures are out of interest (with respect to discussed conformational effects) and can be removed.

  1. 20, line 697: product numeration (4) should be bold.

Ref. 2: correct page numbers to read ‘101-105’.

Ref. 7: shift the dot to read’ Int. J. Mol.’

Author Response

Dear Editor,

thanks for your valued feedback that is aimed at raising the quality of our manuscript. We wholeheartedly accept the recommendations. Your suggestions have been duly and carefully incorporated in the manuscript. We hope that now we have prepared a well-revised manuscript. All comments are colored in text with yellow.

- line 31: the general statement about the type of heteroatom-activity mechanism relationship should be supported with a series of appropriate general reviews rather than a single research article (ref.1)

We have done it.

- lines 49-52: this sentence should be supplemented with a proper citation.

We have supplemented it with a proper citation.

- Introduction: the concept of combining acridine pharmacophore with acylhydrazone unit for applications in biology and material sciences is known. Some recent work deserve mentioning (e.g. Aarjane et al. Arab. J Chem. 2020, 13, 6236; Ma et al. Soft. Matter 2021, 17, 7813)

We have included it in the text.

- chapter 2.4.1.: the decreasing effect of metabolic activity of A549 follows the decreasing stability of sigma bonds in the tested compounds of series 3 (F > H > Cl > Br). This observation should be briefly discussed.

We have briefly discussed this observation.

 Further minor corrections:

Abstract, line 18: remove semicolon to read ‘complex 3b’

Abstract, line 23: compound 3a should be written bold

p.4, Figure 4: for the sake of the readers convenience, the depicted structures should be larger. Also, some hydrogen atoms in these structures are out of interest (with respect to discussed conformational effects) and can be removed.

  1. 20, line 697: product numeration (4) should be bold.

Ref. 2: correct page numbers to read ‘101-105’.

Ref. 7: shift the dot to read’ Int. J. Mol.’

All the changes we have corrected.

Reviewer 2 Report

The scientific article entitled: “Acridine based N-acylhydrazone derivatives as potential anticancer agents: Synthesis, characterization and ctDNA/HAS spectroscopic binding properties” reports the synthesis of acridine N-acylhydrazone derivatives and their potential role of the inhibition of topoisomerase I/II and their binding (ctDNA, HSA).

Moreover, the biological activities of novel acridine-based derivatives as potential anticancer agents reducing the proliferation of A549 and CCD-18Co were also evaluated.

In my opinion this work should be considered for the publication in “Molecules” after revisions.

First, regarding the introduction I would suggest to the authors to stress the importance of heterocyclic rings as a pharmacophore moiety of several molecules with different biological activities, including anticancer and antibacterial. Indeed, the presence of nitrogen heteroatom in the structure improves the interaction with target proteins, enzymes, and receptors through the formation of several type of interaction, such as: hydrogen bonds, dipole-dipole, hydrophobic interactions, van der Waals forces and π-stacking interactions.

At this purpose I suggest to the authors to cite the following recent papers:

  • Di Franco, S. et al., CHK1 inhibitor sensitizes resistant colorectal cancer stem cells to nortopsentin. iScience, 2021, https://doi.org/10.1016/j.isci.2021.102664
  • Vitaku, E., Smith, D. T., & Njardarson, J. T. (2014). Analysis of the structural diversity, substitution patterns, and frequency of nitrogen heterocycles among U.S. FDA approved pharmaceuticals. J Med Chem,57(24), 10257–10274. https://doi.org/10.1021/jm501100b
  • Carbone, A.; et al. Thiazole Analogues of the Marine Alkaloid Nortopsentin as Inhibitors of Bacterial Biofilm Formation. Molecules, 2021, 26, 81. https://doi.org/10.3390/molecules26010081
  • Hamdy, R.; et al. Design, Synthesis and Evaluation of New Bioactive Oxadiazole Derivatives as Anticancer Agents Targeting Bcl-2. Int J Mol Sci, 2020, 21, 8980. https://doi.org/10.3390/ijms21238980
  • Hamdy, R.; et al. “New Bioactive Fused Triazolothiadiazoles as Bcl-2-Targeted Anticancer Agents.” Int J Mol Sci, 2021, 22 12272. https://doi.org/10.3390/ijms222212272

 In the section 2.1 “Chemistry”, I suggest to the authors to report the experimental data such as mmol and number of equivalents only in the section 3 “Material and Methods”.

In the part concerning the biological evaluation, the authors reported the results of the MTT assay for compounds 3a-d in A549 non small cell lung cancer cell line and normal colon fibroblast cell line (CCD-18Co). The IC50 values were calculated for all compounds, and it was observed that in A549 cell line, the compounds did not show a marked cytotoxic effect after 24h, with IC50 higher than 75 µM, except for the derivative 3c, that showed a slight better value of IC50 of 73 µM. In the other hand, after 48h, the IC50 values were in the range of 37-62 µM. Contrarily, a marked toxic effect was observed for the compounds in normal colon fibroblast cell lines CC-18Co with IC50 values in the range of 8–17 µM for compounds 3b-d, while for compound 3a the IC50 value was higher than 75 µM.

These results, clearly demonstrated the toxicity of the novel acridine benzohydrazide analogs in normal human colon fibroblast, limiting clinical advance of this series of new synthesized compounds.

Therefore, from a pharmaceutical chemistry standpoint, these of compounds aren’t particularly interesting for further developments. To reduce the toxicity of these compounds, I suggest to the authors to evaluate some chemical modifications in order to increase the selectivity between normal and cancer cells.

In the paragraph 3.2. “Synthesis of compounds 3a–e, 4, 5 and 7b–d”, though, the authors fully report the NMR spectra data in the supporting information, I suggest to the authors to also report the NMR analysis in the paragraph 3.2.

 In the paragraph 3.2.2. “Synthesis of methyl acridine-4-carboxylate (4)”, in the line 697 please change the font of 4 in bold.

Author Response

Thanks for your valued feedback that is aimed at raising the quality of our manuscript. We wholeheartedly accept the recommendations. Your suggestions have been duly and carefully incorporated into the manuscript. We hope that now we have prepared a well-revised manuscript. All comments are colored in text with purple.

First, regarding the introduction, I would suggest to the authors to stress the importance of heterocyclic rings as a pharmacophore moiety of several molecules with different biological activities, including anticancer and antibacterial. Indeed, the presence of nitrogen heteroatom in the structure improves the interaction with target proteins, enzymes, and receptors through the formation of several type of interaction, such as: hydrogen bonds, dipole-dipole, hydrophobic interactions, van der Waals forces and π-stacking interactions.

For this purpose I suggest to the authors cite the following recent papers - we have used selected papers in the text.

  • Di Franco, S. et al., CHK1 inhibitor sensitizes resistant colorectal cancer stem cells to nortopsentin.iScience, 2021, https://doi.org/10.1016/j.isci.2021.102664
  • Vitaku, E., Smith, D. T., & Njardarson, J. T. (2014). Analysis of the structural diversity, substitution patterns, and frequency of nitrogen heterocycles among U.S. FDA approved pharmaceuticals. J Med Chem,57(24), 10257–10274. https://doi.org/10.1021/jm501100b
  • Carbone, A.; et al. Thiazole Analogues of the Marine Alkaloid Nortopsentin as Inhibitors of Bacterial Biofilm Formation. Molecules, 2021, 26, 81. https://doi.org/10.3390/molecules26010081
  • Hamdy, R.; et al. Design, Synthesis and Evaluation of New Bioactive Oxadiazole Derivatives as Anticancer Agents Targeting Bcl-2. Int J Mol Sci, 2020, 21, 8980. https://doi.org/10.3390/ijms21238980
  • Hamdy, R.; et al. “New Bioactive Fused Triazolothiadiazoles as Bcl-2-Targeted Anticancer Agents.” Int J Mol Sci, 2021, 22 12272. https://doi.org/10.3390/ijms222212272

 In the section 2.1 “Chemistry”, I suggest to the authors to report the experimental data such as mmol and number of equivalents only in the section 3 “Material and Methods”.

We have done it.

In the part concerning the biological evaluation, the authors reported the results of the MTT assay for compounds 3a-d in A549 non small cell lung cancer cell line and normal colon fibroblast cell line (CCD-18Co). The IC50 values were calculated for all compounds, and it was observed that in A549 cell line, the compounds did not show a marked cytotoxic effect after 24h, with IC50 higher than 75 µM, except for the derivative 3c, that showed a slight better value of IC50 of 73 µM. In the other hand, after 48h, the IC50 values were in the range of 37-62 µM. Contrarily, a marked toxic effect was observed for the compounds in normal colon fibroblast cell lines CC-18Co with IC50 values in the range of 8–17 µM for compounds 3b-d, while for compound 3a the IC50 value was higher than 75 µM.

These results, clearly demonstrated the toxicity of the novel acridine benzohydrazide analogs in normal human colon fibroblast, limiting clinical advance of this series of new synthesized compounds. Therefore, from a pharmaceutical chemistry standpoint, these of compounds aren’t particularly interesting for further developments. To reduce the toxicity of these compounds, I suggest to the authors to evaluate some chemical modifications in order to increase the selectivity between normal and cancer cells.

You are right, the results from the MTT assay show that the studied compounds also affected the metabolic activity of healthy cells. However, we have also examined an even more pronounced effect with doxorubicin as positive control and a known chemotherapeutic agent used in the treatment of various types of tumors. As you state, we found the weakest effect on healthy cells only in the case of derivative 3a. Therefore, based on these results, we included in further analyzes only derivative 3a as the substance with the weakest effect on healthy fibroblasts and substance 3c, the effect of which was more pronounced. As can be seen from Figures 9, 10, and 11, for compound 3a a concentration-dependent antiproliferative effect on A549 lung adenocarcinoma cells was also examined. However chemical modifications which led to increasing the selectivity between normal and cancer cells will be the subject of our further studies.

In the paragraph 3.2. “Synthesis of compounds 3a–e, 4, 5 and 7b–d”, though, the authors fully report the NMR spectra data in the supporting information, I suggest to the authors to also report the NMR analysis in the paragraph 3.2.

We have reported it.

 In the paragraph 3.2.2. “Synthesis of methyl acridine-4-carboxylate (4)”, in the line 697 please change the font of 4 in bold.

We have changed it.

Round 2

Reviewer 2 Report

The revised version of scientific article entitled: “Acridine based N-acylhydrazone derivatives as potential anti-cancer agents: Synthesis, characterization and ctDNA/HSA 2 spectroscopic binding properties”, still require further improvements for publishing in “Molecules” and cannot be accepted in the present form.

As already stated in the first revision, is extremely important for a pharmaceutical chemistry standpoint, stress the pharmacological importance of molecules bearing nitrogen heterocycles. In fact, compared to the first version, the authors have implemented the bibliographic references adding only general reviews (references 1-5), however the addition of updated scientific works on nitrogen heterocycle compounds characterized by a marked antiproliferative and antibacterial activity has been omitted.

Therefore, I recommend to the authors to cite the following recent scientific articles:

  • Di Franco, S. et al., CHK1 inhibitor sensitizes resistant colorectal cancer stem cells to nortopsentin. iScience, 2021, https://doi.org/10.1016/j.isci.2021.102664
  • Carbone, A.; et al. Thiazole Analogues of the Marine Alkaloid Nortopsentin as Inhibitors of Bacterial Biofilm Formation. Molecules, 2021, 26, 81. https://doi.org/10.3390/molecules26010081
  • Shao, M. et al., Design, Synthesis, and Biological Evaluation of Aminoindazole Derivatives as Highly Selective Covalent Inhibitors of Wild-Type and Gatekeeper Mutant FGFR4. J Med Chem, 2022, 65, 6, 5113. https://doi.org/10.1021/acs.jmedchem.2c00096
  • Hamdy, R.; et al. Design, Synthesis and Evaluation of New Bioactive Oxadiazole Derivatives as Anticancer Agents Targeting Bcl-2. Int J Mol Sci, 2020, 21, 8980. https://doi.org/10.3390/ijms21238980

Author Response

Good morning,

in the attachment we send the corrected manuscript. We added what the opponent requested.

Best regards

Maria Kozurkova
